# The Association between Types of COVID-19 Information Source and the Avoidance of Child Health Checkups in Japan: Findings from the JACSIS 2021 Study

**DOI:** 10.3390/ijerph19159720

**Published:** 2022-08-07

**Authors:** Masafumi Ojio, Yuto Maeda, Takahiro Tabuchi, Takeo Fujiwara

**Affiliations:** 1Department of Global Health Promotion, Tokyo Medical and Dental University, Tokyo 113-8519, Japan; 2Cancer Control Center, Osaka International Cancer Institute, Osaka 541-8567, Japan

**Keywords:** COVID-19, pandemic, child health checkup, avoidance, information source, fear

## Abstract

The coronavirus disease 2019 (COVID-19) pandemic can affect children’s well-being through mothers’ avoidance of health checkups for children due to media portrayal of the disease. This study investigated the association between the type of information source for COVID-19 received by mothers and the avoidance of their children’s health checkups. The study was an online-based survey, and the participants comprised 5667 postpartum women with children aged under 2 years during the study period. We analyzed the analytic sample and three groups of women with children aged 0–3 months, 4–6 months, and 6 months or older according to the timing of children’s health checkups in Japan. Among the participants, 382 women (6.7%) avoided their children’s health checkups. Multivariate logistic regression analysis revealed that mothers with children over 6 months who used magazines as an information source about COVID-19 tended to avoid their children’s health checkups (adjusted odds ratio (aOR): 3.19; 95% confidence interval (CI): 1.68–6.05) compared with those who did not. In contrast, those using public websites were less likely to avoid their children’s health checkups (aOR 0.58, 95% CI 0.43–0.77). This study showed that specific types of information source on COVID-19 could have varying effects on mothers’ decisions about their children’s health checkups.

## 1. Introduction

Health surveillance and regular health checkups for children play an important role in promoting the healthy development of children. The main purpose of the checkups is to detect diseases at the earliest stage, disability, or household dysfunction, leading to appropriate interventions [1]. In Japan, health checkups for children include the detection of congenital diseases, developmental problems, obesity and tooth decay, and child maltreatment. Local governments are required to provide checkups under Maternal and Child Health Law, and checkups are mostly conducted at 3 or 4 months, 1.5 years, and 3 years of age by the local government [2]. Of these health checkups, 1-month checkups take place at the hospital or clinic where the infant was born, and 3- or 4-month and 1.5-year checkups, which are obligatory under Maternal and Child Health Law, are routinely conducted by public health centers under the local government or clinics [3]. These checkups are free for all children and parents across Japan, and the average percentage of attendance was over 95% before the COVID-19 pandemic [4]. The significance of child health checkups has also been confirmed worldwide. A previous study of 3487 children in Sweden reported that the percentage of suspected new health problems varied between 1.9% and 2.8% at routine health examinations at the 2-, 6-, 9-, and 12-month checkups, and the percentage of minor and moderate problems were from 34% to 44% and from 10% to 15% for newly detected health problems [5]. Another previous literature review on pediatric screening for developmental delay in the US indicated that early health checkups for subtle disabilities such as language impairment, mild intellectual disabilities, and learning disabilities are generally thought to prevent poor health status and school failure [6].

The coronavirus disease 2019 (COVID-19) pandemic has impaired children’s well-being not only through the direct health consequences of viral infection, but also through reduced access to health checkups [7,8]. A previous retrospective study of 935 children who underwent neonatal hip ultrasonography during the COVID-19 pandemic in Turkey reported that the number of late diagnoses of developmental dysplasia of the hip, a congenital condition that should be detected through child health checkups, has increased due to fewer parents applying for checkups [9].

A previous study using semi-structured interviews with caregivers of children in the UK reported that a delay in the decision to access healthcare during the COVID-19 pandemic was influenced by the fear of COVID-19, driven by community perceptions and experiences and by media portrayals [10]. Another previous study that investigated people’s health behavior has shown the differences in the types of media as an information source. While traditional media, such as television and newspapers, often have a key role in providing evidence-based information to the general public [11], social media, a representative of new media, are used more during disasters or emergencies, as they are easy to access and reflect new information quickly [12]. However, social media users can spread information without fact-checking, including inaccurate data and rumors, thereby excessively arousing anxiety and fear [13]. Therefore, the type of media used as an information source may influence people’s health behavior during the COVID-19 pandemic [14]. Actually, previous studies have reported that the types of information source about COVID-19 were associated with health-seeking behaviors on dental care and adherence to COVID-19 infection preventive behaviors [15,16]. However, to the best of our knowledge, there are no studies examining the association between the type of information source of COVID-19 and the avoidance of child health checkups.

In this study, we investigated the association between the type of information source and the avoidance of children’s health checkups using the results from the Japan COVID-19 and Society Internet Survey (JACSIS) [17], a nationwide online-based survey in Japan.

## 2. Materials and Methods

### 2.1. Data Setting

This was a retrospective observational, cross-sectional study using an online-based survey conducted as part of the JACSIS. The study samples for each survey were retrieved from the pooled panels of an internet research agency, Rakuten Insight [18]. As an incentive for responding to the study, participants received “Epoints”, which can be used for online shopping. One Epoint was roughly JPY 100 (approximately USD 0.8 in July 2022). First, a screening survey was conducted for 440,323 registered Japanese women on 24 July 2021 to determine the eligibility of participants whose due date was on or before 31 December 2021, and postpartum women who had delivered a live singleton after July 2019. Among the participants, 2425 pregnant women and 11,661 postpartum women were determined to be eligible for this survey. Next, an internet research agency distributed a questionnaire to all eligible pregnant women through email. Valid responses were obtained from 8047 women (57.1%) between 28 July and 30 August 2021 during the fifth wave of the COVID-19 pandemic in Japan. Although there was no official definition of the fifth wave, the fifth wave is generally defined as the period of the fifth sharp increase in the number of infections from government statistics (from July to September 2021) [19]. Following the exclusion of 1791 women who were pregnant, 6256 postpartum women were included in the analysis. After we determined the participants eligible for the analysis, we excluded women as follows: women who had a child aged over two years (21 women), and women who provided irrelevant or contradictory information (568 women). This method was used in previous studies by the same research group [20,21]. Participants were fully represented from all prefectures in Japan [22] (Appendix A).

### 2.2. Measurements

The primary outcome was to determine the avoidance of child health checkups conducted mainly at 1 month, 3 or 4 months, and 1.5 years of age to check the growth and development of children. The avoidance of child health checkups was determined using the following question: “Did you refrain from having an infant health checkup at a medical institution or a health center for the child born this time?”, with an answer of either “yes” or “no”.

The primary exposure was the types of information source about COVID-19, determined by the following question: “Which of the following do you use as a reliable information source about COVID-19”. Participants were asked to respond with “I use” or “I do not use”, respectively, to each of the following information sources: public websites (websites of government agencies, prefectures, or municipalities), university or scientific-society websites, web news, broadcast media, LINE (a popular social media platform in Japan, similar to WhatsApp), Twitter, Facebook, Instagram, magazines, books, newspapers, TV news, and radio. This classification was developed with reference to the questionnaire used in a previous study that assessed information-seeking behavior for dental care in Italy during the coronavirus pandemic [15].

The covariates were selected from the previous study that examined factors influencing child health checkups using data from a longitudinal household panel survey in Japan [23]. The study indicated that maternal age at birth, birth order, parents’ educational attainment, household income, the frequency of help from grandmothers, and the number of communicating neighbors were factors that influenced health checkup attendance. In the previous study, the number of communicating neighbors was defined as the number of neighbors with whom participants could have daily communication, as a proxy for a structural social network [24,25]. The impact of each factor differed depending on the timing of the checkups. We therefore defined potential risk factors for the avoidance of child health checkups as follows: parity (nulliparous or multiparous); maternal age at delivery (year); child age upon answering; living with grandparents (yes or no); household income in Japanese Yen (JPY) (<JPY 3 million, JPY 3 million to 6 million, JPY 6 million to 10 million, JPY 10 million or more, or unknown); academic background (junior high school graduate, high school graduate, junior college or technical school graduate, university or higher, or unknown); trust for the neighbors (yes, partly, not so much, not at all); number of people to consult (0, 1–2, 3–5, 6–10, 11–15, 16–20, or ≥21); and all the types of information source for COVID-19. We also included prefectures of residence as covariates because the prevalence of COVID-19 infection differed in each area and the fear of COVID-19 infection could influence avoidance [17]. Additionally, we also included parental health-literacy as a factor because low parental health-literacy is thought to affect child healthcare due to an inability to acquire the correct knowledge and act on disease prevention, acute disease care, and chronic disease care [26,27]. To assess parental health-literacy (high or low), we used the scale item constructed in a previous study about finding or using information related to illness or health; participants were asked whether they would be able to “collect information from various sources such as newspaper, book, TV, and the internet”, “extract the information you wanted”, “understand and communicate the obtained information”, “consider the credibility of the information”, and “make decisions about plans or actions to improve your health based on the information” [28]. Each item was rated on a 5-point scale ranging from 1 (strongly disagree) to 5 (strongly agree) to determine their abilities. Participants were categorized into two groups based on the median score of this question (higher health-literacy group, ≥4; lower health-literacy group, <4).

### 2.3. Statistical Analysis

Descriptive statistics were calculated for the analytic sample, including avoidance and non-avoidance groups. *T*-tests were used to compare the averages of continuous variables and chi-square tests to compare the proportions of categorical variables between the groups. We performed multiple logistic regression analysis to calculate the odds ratios (OR) and 95% confidence intervals (CIs) of the information source about COVID-19 for the avoidance of child health checkups after adjustment of the covariates. Parity, maternal age at delivery (year), child age upon answering, living with grandparents, household income, academic background, health literacy, trust for the neighbors, prefectures of residence, and all the types of information source for COVID-19 were considered as covariates. We further conducted these analyses on a total sample and 3 different age groups (less than 3 months, 3–6 months, and over 6 months). Additionally, as a correction for multiple comparisons, the Bonferroni correction was applied, with a significance level of 0.05 divided by the number of analyses. 

The data were analyzed using STATA version 16.0 (StataCorp LLC: College Station, TX, USA). A two-sided *p*-value of less than 0.05 was considered to indicate statistical significance.

## 3. Results

The demographic information of participants, stratified by the avoidance of child health checkups, is shown in Table 1. There were 382 (6.7%) women who avoided health checkups for their youngest child born in the study period. The percentage of women who avoided child health checkups was 4.1% (27/666) among women with a child aged less than 3 months, 4.8% (39/808) among women with a child aged 3–6 months, and 7.5% (316/4193) among women with a child aged over 6 months. There were significant differences between the groups in terms of health literacy and trust for neighbors (*p*-value = 0.002 and *p*-value = 0.007, respectively).

The associations between the avoidance of child health checkups and the information source for COVID-19 are shown in Table 2. Overall, in the total sample, we found that the use of magazines (adjusted odds ratio (aOR): 2.37, 95% confidence interval (CI): 1.31–4.28), university or scientific-society websites (aOR: 1.48, 95% CI: 1.08–2.04), and Twitter (aOR 1.34, 95% CI 1.01–1.77) as information sources significantly increased the risk of avoiding child health checkups, whereas the use of public websites as an information source (aOR: 0.62, 95% CI: 0.48–0.80) prevented avoidance, although all of these were not significant after applying the Bonferroni correction.

In the analysis stratified by child age upon answering, we found two statistically significant results. In the group of children over 6 months, the use of magazines’ information was a significantly high risk factor for the avoidance of health checkups (aOR: 3.19, 95% CI: 1.68–6.05), and the use of public websites’ information was a preventive factor for avoidance (aOR 0.58, 95% CI 0.43–0.77), even after applying the Bonferroni correction (*p* < 0.001). In the same group, the use of university or scientific-society websites’ information was likely to increase the avoidance of health checkups, but not significantly, while the use of other web news’ information was not associated with avoidance. As for social media, the use of broadcast media and Twitter as information sources were likely to be associated with the avoidance of health checkups, although not significantly, whereas the use of LINE, Facebook, and Instagram as information sources were not associated with avoidance.

## 4. Discussion

This study found that the use of magazines as a resource of COVID-19 information was associated with risk of avoiding health checkups for children over 6 months, while the use of public websites’ information was associated with the promotion of health checkups for the same group. The results suggest that understanding the impact of the type of information source on maternal health behavior was crucial for promoting correct health behavior.

During the study period, the cumulative number of infections increased from 892,753 on 28 July to 1,476,805 on 30 August, and the number of new infections reached a previous high of 25,992 (weekly average: 21,247.4) on 20 August during the survey period [22]. In June 2020, the Japanese government urged that children should not delay vaccination and health checkups due to the COVID-19 pandemic through its website, Twitter, and leaflets [29]. The Japan Pediatric Society also provided related information through the Q&A section, promoting children’s health checkups without delay, even during the COVID-19 pandemic, on its website by May 2020 [30]. Despite these efforts, 6.7% of the women analyzed in the current study avoided health checkups for their latest child, which was higher than the pre-pandemic 3- or 4-month and 1.5-year checkup rates in Japan. It has been pointed out that the fear of COVID-19 infection may lead to delays in children’s medical access in previous studies [10,31], and it is possible that the same reason may have caused the avoidance of health checkups.

The types of information source about COVID-19 could have different effects on health behavior during the COVID-19 pandemic. For example, a previous cross-sectional online survey on 3358 participants in Saudi Arabia found that social media, such as Twitter, reduced compliance with infection-preventive behaviors, while government information promoted them [16]. The results of our study were consistent with those of the aforementioned study; Twitter as a source of information on COVID-19 tended to be risky for the avoidance of health checkups in the total sample, while public website showed a significant preventive effect on the avoidance of health checkups. This was due to the traits of social media during a pandemic of infectious disease. Social media was reported as the most accessible source of information for people during the pandemic. However, social media may contain inaccurate information and data that have not been fact-checked [11,12,13,14]. Exposure to COVID-19 misinformation spread on social media is considered to have a negative impact on health behaviors [32]. In contrast, exposure to public websites could have a positive impact on health behaviors, which may be because public websites convey fact-checked information and correct messages, as the Japanese government urged the public to not delay attending health checkups due to the COVID-19 pandemic.

We found that magazines as a COVID-19 information source were significantly associated with the avoidance of health checkups in the group of children over six months. As far as we know, there are few studies that examine the association between magazines and children’s access to health checkups. Many magazines, including those read by parents, are commercial publications and are likely to contain eye-catching articles and advertisements to appeal to a large audience; however, they do not always contain articles written by experts or evidence-based information to help consumers make the desired health decisions. For example, in a previous study of 135 parenting magazine issues in the US, despite the importance of weight gain before and after pregnancy for women’s health, about a quarter of the magazines included at least one feature article on weight loss; moreover, fewer than half of the articles displayed author credentials such as their degree, qualifications, or expertise [33]. Another study of two popular American men’s health-related magazines found that 57.2% of 161 recommendations for men’s health, such as exercise, nutrition, and over-the-counter medication, lacked support from high-quality peer-review scientific evidence. In contrast, 76.4% of the recommendations included unclear, nonexistent, or contradictory evidence [34]. As with social media, magazines do not necessarily contribute to health promotion. On the contrary, inaccurate information may inhibit positive health behaviors.

It should be noted that a statistically significant difference was only obtained in the group of children aged over 6 months. A cross-sectional study of 3032 children who were eligible for 3–4-month, 1.5-year, and 3-year checkups in Japan showed that the rate of not attending checkups increased as the age of the child increased [35]. The study showed that the reason for not attending 3–4-month checkups was mainly that the child had been receiving medical treatment at a medical institution, and that the reasons for not attending 1.5-year and 3-year checkups were the parents’ work schedules or the child’s attendance to a daycare or preschool. Furthermore, many parents whose children did not attend health checkups had problems with their finances, parenting skills, and understanding, and more than 60% of the parents did not show any intention to attend checkups in response to a recommendation. This previous study suggested that as age increased, parents’ awareness of medical checkups declined and led to a lower rate of checkups, and the results of the present analysis may also be because the parents in the group of children aged over 6 months were more susceptible to the influence of media information about COVID-19.

This study has several limitations. First, reverse-causation is likely due to the cross-sectional design of the study. That is, it is possible that women who avoided their children’s health checkups tended to read magazines. Further research is needed to clarify the causation between the source of COVID-19 information and the avoidance of child health checkups. Second, online surveys are generally known to be subjected to selection bias, which limits responses to those who have access to the internet [36]. However, this effect could be considered to be limited because pregnant women make up a relatively young segment of the general population, and the population without access to the internet would be small. Third, we could not identify what kinds of articles or magazines were associated with avoiding children’s health checkups. Finally, although we selected the covariates from a previous similar study [23], unmeasured confounders, such as the availability of childcare services, parents’ working hours, and the number of paid days off work, might have caused a bias in the association.

## 5. Conclusions

In conclusion, magazines as a COVID-19 information resource were found to be a risk for avoiding health checkups among parents with children aged over 6 months, while public websites were preventive. During a pandemic of infectious disease, concerns about non-fact-checked social media have attracted much attention, but there is not sufficient evidence on the impact of magazines. Further research on the types of information source used during a pandemic is warranted to appropriately promote health behaviors.

## Figures and Tables

**Table 1 ijerph-19-09720-t001:** Demographics stratified by avoidance of health checkups for children.

Variables	Total (*n* = 5667)	Avoidance of Child Checkups (−) (*n* = 5285)	Avoidance of Child Checkups (+) (*n* = 382)	*p*-Value
	Mean (SD) or *N* (%)			
**Child age upon answering**				0.003
Less than 3 months	666 (11.7)	639 (12.1)	27 (7.1)	
3–6 months	808 (14.3)	769 (14.5)	39 (10.2)	
Over 6 months	4193 (74.0)	3877 (73.4)	316 (82.7)	
Gestational age at delivery (weeks)	38.7 (2.0)	38.7 (1.9)	38.4 (2.5)	0.005
Nulliparity	3056 (53.9)	2849 (53.9)	207 (54.2)	0.92
Maternal age at delivery (year)	32.2 (4.4)	32.2 (4.4)	31.9 (4.5)	0.14
Living with Grandparents	267 (4.7)	251 (4.8)	16 (4.2)	0.62
**Houseohld Income**				0.16
<3 million	183 (3.2)	166 (3.1)	17 (4.5)	
3 million to 6 million	1614 (28.5)	1489 (28.2)	125 (32.7)	
6 million to 10 million	2252 (39.7)	2112 (40.0)	140 (36.7)	
10 million or more	750 (13.2)	701 (13.3)	49 (12.8)	
Unknown	868 (15.3)	817 (15.5)	51 (13.4)	
**Academic background**				0.40
Junior high school graduate	30 (0.5)	29 (0.6)	1 (0.3)	
High school graduate	872 (15.4)	818 (15.5)	54 (14.1)	
Junior college or technical school graduate	1789 (31.6)	1655 (31.3)	134 (35.1)	
University or higher	2959 (52.2)	2766 (52.3)	193 (50.5)	
Unknown	17 (0.3)	17 (0.3)	0 (0.0)	
**Health Literacy ***				0.002
Low	3523 (62.2)	3257 (61.6)	266 (69.6)	
High	2144 (37.8)	2028 (38.4)	116 (30.4)	
**Trust for neighbors**				0.007
Yes	740 (13.1)	688 (13.0)	52 (13.6)	
Partly	3371 (59.5)	3164 (59.9)	207 (54.2)	
Not so much	1266 (22.3)	1176 (22.3)	90 (23.6)	
Not at all	290 (5.1)	257 (4.9)	33 (8.6)	
**Number of people to consult**				0.08
0	77 (1.4)	72 (1.4)	5 (1.3)	
1–2	2109 (37.2)	1943 (36.8)	166 (43.5)	
3–5	2786 (49.2)	2625 (49.7)	161 (42.2)	
6–10	614 (10.8)	570 (10.8)	44 (11.5)	
11–15	46 (0.8)	44 (0.8)	2 (0.5)	
16–20	11 (0.2)	9 (0.2)	2 (0.5)	
≥21	24 (0.4)	22 (0.4)	2 (0.5)	

Note: *T*-tests were used to compare the averages of continuous variables and chi-square tests to compare the proportions of categorical variables between the groups. A two-sided *p*-value of less than 0.05 was considered statistically significant. * High literacy was defined as an average score of 4 or more on five questions.

**Table 2 ijerph-19-09720-t002:** Association between avoidance of child checkups and information source for COVID-19.

Variables	Crude OR (95% CI)	Adjusted OR (95% CI) ^†^			
		Total	Less than 3 Months (*n* = 666 (11.7%))	3–6 Months (*n* = 808 (14.3%))	Over 6 Months (*n* = 4193 (74.0%))
**Traditional Media**					
Magazine	2.97 (1.92–4.58) *	2.37 (1.31–4.28) *	– ^‡^	2.49 (0.23–26.69)	3.19 (1.68–6.05) **
Book	1.88 (1.12–3.15) *	0.88 (0.45–1.74)	44.47 (2.62–754.64) *	9.56 (1.36–66.97) *	0.49 (0.22–1.12)
Newspaper	1.20 (0.91–1.57)	1.10 (0.79–1.53)	0.71 (0.12–4.41)	1.55 (0.43–5.55)	1.13 (0.78–1.62)
TV news	0.84 (0.68–1.05)	0.88 (0.68–1.14)	0.69 (0.21–2.33)	0.39 (0.15–1.06)	0.94 (0.70–1.25)
Radio	1.89 (1.24–2.88) *	1.50 (0.92–2.46)	22.63 (0.71–721.49)	2.05 (0.24–2.17)	1.74 (1.03–2.95) *
**Online Media**					
Public website	0.71 (0.58–0.89)	0.62 (0.48–0.80)	0.77 (0.21–2.82)	2.05 (0.24–17.88)	0.58 (0.43–0.77) **
University or scientific-society website	1.30 (1.00–1.71) *	1.48 (1.08–2.04) *	5.62 (1.18–26.73) *	0.89 (0.37–2.17)	1.61 (1.13–2.30) *
Web news	0.91 (0.73–1.12)	0.99 (0.77–1.27)	0.43 (0.12–1.46)	0.55 (0.14–2.15)	1.03 (0.78–1.36)
**Social Media**					
Broadcast media (YouTube, TikTok, etc.)	1.59 (1.19–2.12) *	1.38 (0.98–1.93)	0.08 (0.01–1.00) *	0.96 (0.38–2.40)	1.49 (1.01–2.18) *
LINE	1.10 (0.82–1.46)	0.81 (0.57–1.15)	1.18 (0.20–6.83)	1.98 (0.54–7.35)	0.81 (0.55–1.19)
Twitter	1.36 (1.08–1.72) *	1.34 (1.01–1.77) *	0.75 (0.17–3.38)	0.44 (0.11–1.82)	1.38 (1.00–1.89) *
Facebook	1.59 (1.12–2.26) *	1.23 (0.79–1.93)	0.51 (0.04–6.73)	2.29 (0.81–6.52)	1.48 (0.90–2.42)
Instagram	1.11 (0.87–1.42)	0.88 (0.64–1.20)	4.34 (1.01–18.64) *	1.07 (0.35–3.33)	0.71 (0.50–1.03)

Note: Multiple logistic regression analysis was performed to calculate the OR and 95% CIs of information sources about COVID-19 for the avoidance of child health checkups after the adjustment of covariates. Bonferroni correction (with the significance level set at 0.05 divided by the number of analyses to compensate for multiple comparisons) was applied. COVID-19: Coronavirus disease 2019, OR: Odds ratio, CI: confidence interval. ^†^ Covariates included parity, maternal age at delivery, child age, living with grandparents, academic background, household income, health literacy, trust for neighbors, number of people to consult, prefecture of residence, and all types of information source for COVID-19. ^‡^ No women used magazine as an information source of COVID-19. * *p*-value < 0.05. ** *p*-value < 0.001.

## Data Availability

The data that support the findings of this study are available upon reasonable request. However, restrictions apply to the availability of this data because data associated with personal identification are not shared. If any person wishes to verify our data, they are most welcome to contact the corresponding author.

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
