# Peer review of "The Association between Types of COVID-19 Information Source and the Avoidance of Child Health Checkups in Japan: Findings from the JACSIS 2021 Study"

_ijerph, 2022, doi:10.3390/ijerph19159720_

Round 1

Reviewer 1 Report

 This study addresses the important issue of the impact of the COVID-19 pandemic on regular check-ups for young children. The authors found an association between the types of information sources that parents used for information on COVID-19 and regular well-baby check-ups.  There are several areas that the authors need to address, including careful attention to grammar, spelling and sentence structure.

Here are detailed comments to guide your revision:

This study did not analyze the whole population but rather a sample (5,667 participants) (line 16).

Please provide citations for the listed purpose of checkups (line 29/30).

You discuss one study from Sweden but if there are others that warrant the statement of worldwide significance please provide the citation(s) (line 36).

Please rewrite the awkward sentence structure (lines 52-58).

Social media itself does not spread information but rather users of social media (line 57).

Please provide citation for statement that social media can arouse fear and anxiety (line 58).

Sentence structure and grammar needs to be addressed (lines 64-65).

What internet research agency did you use? Can you provide more information? How did they have access to this population? Did they administer the JACSIS?

What was the inclusion criteria that you used (line 70) – what were potential participants screened for?

The dates listed for screening, inclusion criteria, and recruitment are somewhat confusing for the reader. Can you please clarify this?

Exclusion criteria should be moved up to where you discuss screening.

I am not sure what the statement in lines 81-82 is saying. Were there participants from every prefecture?

This section that discusses child health checkups is helpful for the reader and should be moved up to the introduction (line 86-91).

I’m not sure what the statement ‘primary exposure’ refers to. COVID? (line 95).

The background information about type of information and health-seeking behaviors belongs in the introduction (line 96-98).

The grammar in the sentence at line 100 is incorrect. (e.g. change to Participant were asked to respond…).

What do you mean by ‘number of communicating neighbors’ (line 110)? What is the rationale for including this?

Please provide the measure used or citation for trust for the neighbors (line 117). What is the rationale for using this as a co-variate.

I think you are referring to infection rate not status (line 120). Can you please say more about why this may affect child check-up attendance?

The section on the rationale for including parental health literacy, the health literacy measure, and how it was scored is very good.

Again, you are only analyzing the sample not the overall population (line 135).

Please re-word the awkward sentence and poor grammatical structure at line 144-146.

I am confused about your age groups. Were the three groups 0-2 months or up to 3 months? What about children between 2 and 3 months? Please check spacing in line 144.

I think you are referring to demographics not background information (line 150).

Is public website singular or plural (line 167-8)?

What is LINE (p 174)?

Please check spelling, spacing, and format in Table 1 and 2.

It would not be accurate to say risk factor but rather that they are associated with (line 172).

Good results statement (line 181).

All information from the government or from government websites (line 179)?

The brief description of when the study was conducted would be good to have in the methods section earlier (line 183).

Please write out Q & A (line 190).

Please check the wording and sentence structure for the paragraph starting at line 196. (e.g. magazines themselves do not have an impact)

I am not sure what the discussion of magazine content (lines 202-210) offers. The examples you give (lack of agreement as to causes, lack of content, no authority) do not back up your statement that “magazines do not always contain the correct information” (line 209-210). Please provide citations backing up a statement such as this.

Please fix grammar and sentence structure in paragraph starting line 213.

Please provide citation for last statement in line 225-226.

Can you tell me more about your statement regarding your third limitation?

Please provide citation for the sentence line 235-6.

Reviewer 2 Report

The article "The association between COVID-19 information source types and prevention of child health check-ups in Japan: Findings from The JACSIS 2021 study" addresses a very relevant and current topic.
The authors set out to study the association between the type of information source and avoidance of children's health examinations. However, it seems to us that throughout the introduction they could better explain the relevance of the topic based on studies already carried out on sources of information and avoidance of children's health check-ups.
 The methodology deserves a better organisation. At a certain point, the authors choose to indicate a selection of covariates that are not described in the introduction or indicated in the objectives. They even present the table with the covariates without mentioning them in the results and discussion, which leads me to question the reason for their introduction.
The discussion returns to the main theme and can be a little more developed.
Overall I think that this article would benefit from a better general organisation.

Round 2

Reviewer 1 Report

Thank you for your careful revision and edits of your manuscript. There are many areas that have been markedly improved and language clarified. There are only a few minor changes that need to be made after which I recommend this paper's acceptance.

1. [Line 135-136] You provided a clear explanation to me about what you mean by 'number of communicating neighbors'. Please include this in your manuscript as it will help your readers understand what the covariate measures and your rationale for using it (you should not assume that the reader has already read the previous study).

2. [Line 142-143] This should read 'Descriptive statistics were calculated for the entire sample including avoidance and non-avoidance groups'.

3. [Line 173-175] It is not magazines (please make this into plural) or public websites that are factors but rather the use of magazines as an information source/ use of public websites as an information source).

4. [Line 184-187] Again, it is the use of public websites' information. Please clarify this.

5. [Lines 83-85] Please list these efforts chronologically. 

6. [Lines 83-85] How was this determined to be the fifth wave of COVID? Is there a citation you can provide for the reader that may not be as familiar with the COVID pandemic timeframes in Japan.

7. [Lines 219-2022] See comment above about use of magazines as an information source. 

Reviewer 2 Report

The changes made answer most of the questions raised in the first review. The article is much clearer.

Author Response

Thank you very much for reviewing our revised manuscript.

Owing to your insightful comments, we could improve the quality of the manuscript.

We are grateful for your review and advice.